# DDX5 promotes oncogene C3 and FABP1 expressions and drives intestinal inflammation and tumorigenesis

Nazia Abbasi[1],*, Tianyun Long[1],*, Yuxin Li[1], Brian A Yee[1], Benjamin S Cho[1], Juan E Hernandez[1], Evelyn Ma[1], Parth R Patel[1], Debashis Sahoo[4], Ibrahim M Sayed[2], Nissi Varki[2], Soumita Das[2], Pradipta Ghosh[1,3], Gene W Yeo[1], Wendy Jia Men Huang[1]

**Tumorigenesis in different segments of the intestinal tract involves tissue-specific oncogenic drivers. In the colon, complement component 3 (C3) activation is a major contributor to inflammation and malignancies. By contrast, tumorigenesis in the small intestine involves fatty acid–binding protein 1 (FABP1). However, little is known of the upstream mechanisms driving their expressions in different segments of the intestinal tract. Here, we report that the RNA-binding protein DDX5 binds to the mRNA transcripts of *C3* and *Fabp1* to augment their expressions posttranscriptionally. Knocking out DDX5 in epithelial cells protected mice from intestinal tumorigenesis and dextran sodium sulfate (DSS)–induced colitis. Identification of DDX5 as a common upstream regulator of tissue-specific oncogenic molecules provides an excellent therapeutic target for intestinal diseases.**

## Introduction

Tissue-specific oncogenic molecules drive tumorigenesis in different segments of the intestinal tract. In the colon, complement component C3 protein induces the expression of pro-inflammatory cytokines, such as IL-1$\beta$ and IL-17 (1, 2, 3). Ablation of C3 genetically protects against colitis and tumorigenesis in mouse models (4, 5, 6, 7). In the small intestine, fatty acid–binding protein 1 (FABP1) is critical for intestinal absorption of dietary long-chain fatty acids (8, 9). Ablation of FABP1 genetically protects against tumorigenesis in the small intestine (10).

Regulations of C3 and FABP1 expression at the transcriptional level are described in previous reports. *C3* transcription is controlled by the twist basic helix–loop–helix transcription factor 1 (TWIST1), CCAAT/enhancer-binding protein $\beta$ (C/EBP$\beta$), nuclear receptors farnesoid X receptor, and peroxisome proliferator-activated receptor $\alpha$ in response to stimulation from pro-inflammatory cytokines, such as TNF$\alpha$, IFN$\gamma$, and IL1$\beta$ (11, 12, 13, 14, 15, 16, 17). *Fabp1* transcription is controlled by GATA-binding protein 4 (GATA4), C/EBP, peroxisome proliferator-activated receptor $\alpha$, pancreatic and duodenal homeobox 1 (PDX1), and hypoxia-inducible factor (HIF1$\alpha$) (18, 19, 20, 21, 22). However, little is known about how C3 and FABP1 expressions are regulated posttranscriptionally in intestinal epithelial cells (IECs).

Posttranscriptional regulation of gene products can be orchestrated, in part, by RNA-binding proteins (23). One member of the DEAD-box containing RNA-binding protein family, DDX5, is abundantly expressed in the intestinal epithelium (24). Mutation and overexpression of DDX5 are found in human cancers, and its overexpression predicts advanced clinical stage and poor survival in colorectal cancer (CRC) patients (25, 26, 27). Knockdown of DDX5 inhibited the proliferation of cancer cells in vitro and the growth of xenografts in immunodeficient hosts (28, 29).

Mechanistically, DDX proteins have two major modes of action. First, they can directly bind to specific RNA substrates, use ATP hydrolysis energy to unwind RNA duplexes, facilitate RNA annealing, and/or organize RNA–protein complex assembly (30, 31, 32, 33). Second, DDXs can partner with transcription factors to modulate gene transcription (24, 30, 34, 35, 36, 37, 38, 39, 40). In human cancer cell lines, DDX5 interacts with $\beta$-catenin protein and the long non-coding RNA NEAT1 to promote oncogene expression (41, 42). However, we know little about how the RNA-binding properties of DDX5 contribute to shaping the epithelial RNA regulome during homeostasis and tumorigenesis in vivo.

Here, we revealed that DDX5 binds to *C3* and *Fabp1* mRNA and promotes their expressions in primary IECs from the colon and small intestine, respectively. Loss of DDX5 expression in IECs protects against colonic and small intestine tumorigenesis in vivo. Identification of DDX5 as a common upstream regulator of tissue-specific oncogenic molecules provides an excellent therapeutic target for treating intestinal cancers.

[1]Department of Cellular and Molecular Medicine, University of California San Diego, La Jolla, CA, USA   [2]Department of Pathology, University of California San Diego, La Jolla, CA, USA   [3]Department of Medicine, University of California San Diego, La Jolla, CA, USA   [4]Department of Pediatrics, University of California San Diego, La Jolla, CA, USA

Correspondence: wjh003@health.ucsd.edu
*Nazia Abbasi and Tianyun Long contributed equally to this work

# Results

## DDX5 regulates the epithelial immune response program and contributes to inflammation in the colon

In the IECs isolated from the colon and small intestine of adult wild-type (WT) mice, mRNAs encoding 35 RNA-binding DEAD-box containing proteins (DDXs) were found at various levels (Fig S1A and B and Table S1). Among these, *Ddx5* was the most abundant transcript (Fig 1A). Western blot analyses confirmed that DDX5 proteins were present throughout the intestinal tract (Fig 1B). In the colon, immunohistochemistry (IHC) and nuclear-cytoplasmic fraction revealed that DDX5 proteins predominantly localized to the nucleus of IECs (Figs 1C and S1C). Therefore, we hypothesize that DDX5 may bind to target colonic IEC RNAs in the nucleus and regulate their expressions posttranscriptionally.

Hence, we generated an epithelial DDX5 knockout mice (DDX5$^{\Delta IEC}$) using the Villin1 (*Vil1*)–Cre recombination system (Fig S2A). WT$^{IEC}$ and DDX5$^{\Delta IEC}$ littermates were born in Mendelian ratios and had similar growth curves (Fig S2B). IECs isolated from different segments of the intestinal tract confirmed efficient knockout of DDX5 at the RNA and protein levels throughout the small intestine and colon (Fig S2C and D). Comparison of the RNA profile of colonic IECs isolated from steady-state WT$^{IEC}$ and DDX5$^{\Delta IEC}$ mice revealed that knocking out DDX5 resulted in a down-regulation of 306 and up-regulation of 174 colonic IEC transcripts (Fig 1D and Table S2). DDX5-dependent RNA programs of the colonic IECs were enriched with genes involved in immune response activation (Fig 1E and Table S3).

Therefore, we hypothesized that DDX5$^{\Delta IEC}$ mice with reduced immune activation in the colon may be protected against intestinal inflammation during colitis. To test this possibility, we challenged WT and DDX5$^{\Delta IEC}$ mice with 2% DSS in drinking water. By day 9, DDX5$^{\Delta IEC}$ animals experienced less weight loss and recovered more quickly than their WT cohoused littermates (Fig 1F). Colons from DSS-challenged DDX5$^{\Delta IEC}$ animals were longer (Fig 1G) and showed milder histological pathology, particularly in matrices scoring for immune infiltrate, submucosal inflammation, and abnormal crypt density (Fig S3A–C). Furthermore, lamina propria cells from DDX5$^{\Delta IEC}$ mice expressed less transcripts of inflammatory cytokines, including *Il1b* and *Tnf* (Fig S3D).

In humans, colonic tissues from ulcerative colitis (UC) patients have higher DDX5 expression than healthy controls (Fig S4A) (42). Moreover, reduction of DDX5 positively correlates with UC patients responding favorably to anti-TNF therapy (Fig S4B) (43). Together, these results indicate a conserved and unappreciated role of epithelial DDX5 in intestinal inflammation in vivo.

## Epithelial DDX5 promotes colonic tumorigenesis

Excess inflammation, such as those found in inflammatory bowel diseases, predisposes patients to epithelial dysplasia and cancer (44, 45). Elevated expression of DDX5 predicts worse relapse-free survival in CRC patients (25, 26, 27). To assess the contribution of DDX5 to colonic tumorigenesis in vivo, we crossed the DDX5$^{flox}$ line to the adenomatous polyposis coli (*Apc*) mutant mice (*Apc*$^{fl/+}$*Cdx2*Cre$^+$, also known as APC$^{\Delta cIEC}$) (Fig 2A). Previous studies demonstrated that intestinal tumorigenesis in the *Apc* mutant mice is driven by colonic

immune cell–mediated inflammation (44). Haploinsufficiency of the tumor suppressor APC leads to aberrant β-catenin activation and the development of large colonic adenomas (46, 47). In this model, the loss of one copy of the *Apc* allele in the epithelium results in spontaneous tumors, anal prolapse, and subsequent weight loss between day 100 and 120 (44, 48, 49). Periodic acid-Schiff–stained histological sections of colonic adenomas from the APC$^{\Delta cIEC}$ mice showed a loss of differentiated goblet cell population and neoplastic cell infiltration beyond the basal membranes (Fig S5A). Western blot analyses revealed that DDX5 proteins were expressed at a significantly higher level in colonic tumors from APC$^{\Delta cIEC}$ mice than adjacent normal tissues or IECs from non–tumor-bearing WT mice (Fig S5B), similar to findings previously reported in human CRCs (25). At 4 mo of age, APC$^{\Delta cIEC}$ DDX5$^{\Delta cIEC}$ mice had lower incidence of anal prolapse (Fig 2B) and experienced less weight change compared to APC$^{\Delta cIEC}$ DDX5$^{WT}$ controls (Fig 2C). In the colon, APC$^{\Delta cIEC}$ DDX5$^{\Delta cIEC}$ mice had fewer macroscopic tumors (Fig 2D). IHC studies confirmed that DDX5 was indeed knocked out and lesions from the APC$^{\Delta cIEC}$DDX5$^{\Delta cIEC}$ mice had reduced expression of the cell proliferation marker, Ki67 (Fig S5C). However, no significant difference of tumor sizes was found on day 120 (Fig 2E).

We hypothesized that DDX5 contributes to CRC by regulating specific RNA programs in colonic IECs. Consistent with this possibility, Kaplan–Meier analysis of alive and disease-free survival in two independent cohorts and progression-free survival of a third patient cohort indeed reveal strong associations between the DDX5-associated 20 down-regulated gene signature we identified in our colonic IEC RNAseq study and worse CRC outcome (Fig 2F). Together, these results demonstrate that DDX5 is a critical contributor to colonic tumorigenesis in vivo.

## Epithelial DDX5 directly binds *C3* mRNA and enhances its expression posttranscriptionally

To define the direct target of DDX5 in colonic IECs, we performed the enhanced cross-linked immunoprecipitation (eCLIPseq) assay using the anti-DDX5 antibodies (Table S4). Successful pull-down of DDX5 proteins were confirmed by western assays (Fig S6A). Sequencing results were processed by the ENCODE eCLIPseq analysis pipeline, as described in reference 50 and outlined in Fig S6B. Using a cutoff of three in both $\log_{10}$ P-values and $\log_2$ fold changes of immunoprecipitation (IP) signal over input, we identified 201 colonic IEC RNA sites, corresponding to 138 transcripts, that were significantly enriched by the anti-DDX5 antibodies (Table S5). More than 44% of the DDX5-bound sites localized to coding regions on colonic IEC RNAs (Fig S6C). Of the 138 DDX5-bound transcripts, RNA levels of *C3*, *Ahcyl1*, and *Shroom3* were significantly altered in DDX5-deficient colonic IECs (Fig 3A). Notably, the phenotype of the *Apc*$^{mut}$ *C3*-deficient mice (4, 5, 6, 7) mirrored those we observed here in the APC$^{\Delta cIEC}$ DDX5$^{\Delta cIEC}$ mice. Two independent studies in human CRC patients revealed that higher expression of *C3* predicts poor overall and relapse-free survival (47, 48).

*C3* mRNA is the highest expressed member of the complement family in wild-type mouse colonic IECs (Fig S7A). Reduced *C3* transcripts and proteins were found in DDX5-deficient colonic IECs (Fig 3B–D). If DDX5 promotes *C3* mRNA expression at the transcription level, we expect to observe altered RNA polymerase II

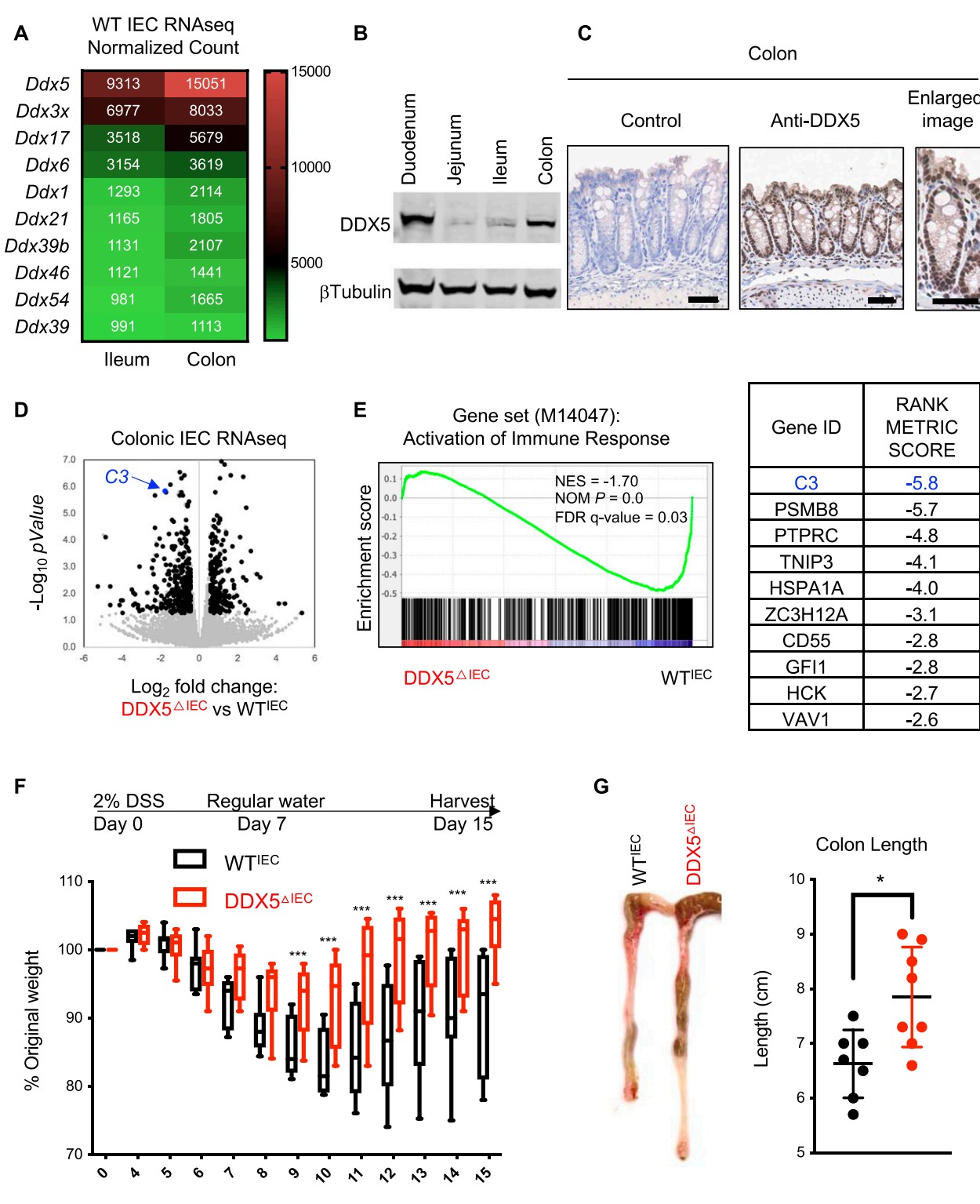

**Figure 1. DDX5 regulates colonic epithelial immune response program and contributes to colitis.**
**(A)** Heat map of average normalized RNAseq read counts of the 10 highest expressed members of the DDX family in the ileum and colon of steady-state WT mice (n = 2).
**(B)** Representative Western blots showing DDX5 and β-tubulin protein expression in intestinal epithelial cells (IECs) from different sections of the intestine in WT mice. Experiments were repeated three times using independent biological samples with similar results. **(C)** Representative images from immunohistochemistry analysis of DDX5 in the colon of WT mice. Enlarged image is shown on the right. Scale bar represents 50 μm. **(D)** Scatterplot of log₂ (fold changes: DDX5$^{ΔIEC}$ over WT$^{IEC}$) and −log₁₀(P-values) of colonic IEC transcripts. RNAseq was performed on two independent pairs of cohoused DDX5$^{ΔIEC}$ over WT$^{IEC}$ littermates. Black dot: DDX5-dependent transcripts

recruitment and deposition of H3 lysine 4 trimethylation (H3K4me3) on the *C3* gene promoter in colonic IECs from DDX5$^{\Delta IEC}$ mice. However, chromatin immunoprecipitation (ChIP) qRT-PCR assay showed that a similar enrichment of RNA polymerase II and H3K4me3 were found on the *C3* promoter in colonic IECs from WT$^{IEC}$ and DDX5$^{\Delta IEC}$ mice (Fig S7B). In addition, fractionation studies revealed that C3 level is similar in the nuclear compartment but significantly reduced in the cytoplasm of DDX5-deficient IECs (Fig S7C). Next, we asked whether DDX5 regulation of *C3* is intrinsic to colonic epithelial cells and independent of inputs from gut microbiota and immune cells using an organoid culture system. Briefly, colonic crypts containing epithelial stem cells were harvested from WT$^{IEC}$ and DDX5$^{\Delta IEC}$ littermates and maintained ex vivo for 5–7 passages. RNAseq of these colonic organoids revealed a similar reduction of *C3* RNA in DDX5-deficient cultured cells (Fig S7D), suggesting that the regulation of *C3* by DDX5 is epithelial cell intrinsic.

Results from the eCLIPseq assay revealed that DDX5 was enriched on a region of the *C3* transcript encoded by exon 30 (Fig 3E). Therefore, we hypothesize that DDX5 may bind to and regulate *C3* transcripts at the posttranscriptional level. Insertion of the short stretch of DDX5-bound region of mouse *C3* into the 3′UTR of the psiCheck2 reporter was sufficient to potentiate DDX5-dependent *Renilla* luciferase activity in a human epithelial cell line (Fig 3F). In flavipiridol-treated human epithelial cells, *C3* mRNAs experienced a greater turnover when DDX5 was knocked down (Fig 3G). Together, these results suggest that DDX5 binds to and promotes *C3* mRNA stability in colonic IECs.

### Epithelial DDX5 promotes small intestine tumorigenesis

In the wild-type mouse small intestine, DDX5 is also abundantly expressed under steady state (Figs 1A and 4A). To ask whether DDX5 may also be involved in tumorigenesis of the small intestine, epithelial DDX5 conditional mice (*Ddx5*$^{flox}$) were crossed to the *Apc*$^{fl/+}$*Vil1Cre*$^+$ mice (Fig 4B). Different from the *Apc*$^{fl/+}$*Cdx2Cre*$^+$ mutant mice described in Fig 2, APC$^{\Delta IEC}$ mice harbor intestinal tumors in both the small intestine and colon. APC$^{\Delta IEC}$ mice begin to experience significant weight loss starting around day 100 of age. By 110 d of age, APC$^{\Delta IEC}$DDX5$^{\Delta IEC}$ littermates continued to gain weight, but APC$^{\Delta IEC}$ DDX5$^{WT}$ mice began to experience significant weight loss. By day 120, APC$^{\Delta IEC}$ DDX5$^{WT}$ and APC$^{\Delta IEC}$ DDX5$^{\Delta IEC}$ mice had significant weight differences (Fig 4C). Macroscopic tumor numbers in the jejunum, ileum, and colon of the APC$^{\Delta IEC}$ DDX5$^{\Delta IEC}$ mice were significantly lower than those found in the APC$^{\Delta IEC}$ DDX5$^{WT}$ mice (Fig 4D and E). No statistical significance was observed in tumor numbers found in the duodenum. The average tumor sizes were comparable between WT and DDX5-deficient tissues (Fig 4F and G), consistent with the results observed in the *Apc*$^{fl/+}$*Cdx2Cre*$^+$ mutant mice (Fig 2E). These results uncover a novel role of epithelial DDX5 in promoting small intestinal tumorigenesis.

### DDX5 binds a distinct set of RNAs to drive tumorigenesis in the small intestine

Next, we asked whether DDX5 regulates overlapping and/or distinct RNA programs in the small intestine and colon. As most DDX5-dependent tumorigenesis of the small intestine occurred in the distal end (Fig 4E), we focused on characterizing the DDX5-dependent RNA program in the ileal section of the small intestine. Global transcriptome analyses revealed that DDX5 controls overlapping and distinct programs in the ileum and colon (Fig 5A and Table S6).

To determine the direct targets of DDX5 in the small intestine, eCLIPseq was performed using UV cross-linked cells from WT mice (Fig S8A). Overall, we found DDX5 binding to 1,276 small intestine IEC RNA sites, corresponding to 466 transcripts (Fig 5B and Table S7). Similar to colonic IECs (Fig S6C), DDX5 was also enriched on coding regions of small intestine IECs (Fig S8B). Of all the DDX5-bound small intestine transcripts, seven experienced significant altered RNA expression in DDX5-deficient ileal IECs (Fig S9A). Increased expression of *Fabp1*, but not the others, significantly correlates with worse relapse-free survival in CRC patients (Fig S9B).

*Fabp1* encodes FABP1 and is uniquely found in the small intestine IECs (Fig S10A), consistent with previous reports (51, 52). Knocking out FABP1 in mice protects against small intestine tumorigenesis (10), which phenocopied our observations in the DDX5$^{\Delta IEC}$ mice. On the *Fabp1* RNA, DDX5 localized to a region encoded by exon 2 (Fig 5C). *Fabp1* mRNA and its protein were significantly reduced in DDX5-deficient small intestine IECs (Figs 5D and E and S10B). Transcripts coding for other members of the FABP family were DDX5 independent, suggesting a unique regulation of *Fabp1* by DDX5. Similar abundance of mature *Fabp1* mRNAs was found in the nucleus of WT$^{IEC}$ and DDX5$^{\Delta IEC}$ IECs, but cytoplasmic mature *Fabp1* mRNAs were significantly lowered in the DDX5$^{\Delta IEC}$ IECs (Fig 5F). These results suggest that DDX5 binds to and promotes *Fabp1* mRNA stability in small intestine IECs. Last, we asked whether binding of DDX5 to *Fabp1* mRNAs in the small intestine IECs may also affect ribosome recruitment for protein translation. We found that ribosomal engagement of *Fabp1* mRNA in the small intestine was significantly decreased in cells from DDX5$^{\Delta IEC}$ mice (Fig 5G). Together, these results reveal that DDX5 regulates unique IEC targets through overlapping and distinct posttranscriptional mechanisms (modeled in Fig 5H).

## Discussion

CRC is the fourth most deadly cancer worldwide (53), where DDX5 is often mutated and/or overexpressed (54). The higher expression of DDX5 predicts poor patient survival (25, 26, 27). Here, we

---

defined as log$_2$ (fold changes: DDX5$^{\Delta IEC}$ over WT$^{IEC}$) ≥0.5 or ≤−0.5 and *P*-value < 0.05 (DESeq). *C3* is indicated in blue. **(E)** Left: Gene set enrichment analysis of immune response activation (M14047) in DDX5-deficient and DDX5-expressing colonic IECs from steady-state mice. NES, normalized enrichment score; NOM *P*, normalized *P*-value. Right: Ranked top 10 DDX5-regulated genes involved in immune response activation. **(F)** Weight loss of WT$^{IEC}$ (n = 7) and DDX5$^{\Delta IEC}$ (n = 9) mice challenged with 2% DSS in their drinking water. This experiment was repeated twice with similar results. Error bars represent SD. ***$P$ < 0.001 (multiple *t* test). **(G)** Colonic length in mice from (F) on day 15 post-DSS challenge. Each dot represents one mouse. Results are means ± SD. *$P$ < 0.05 (*t* test).
Source data are available for this figure.

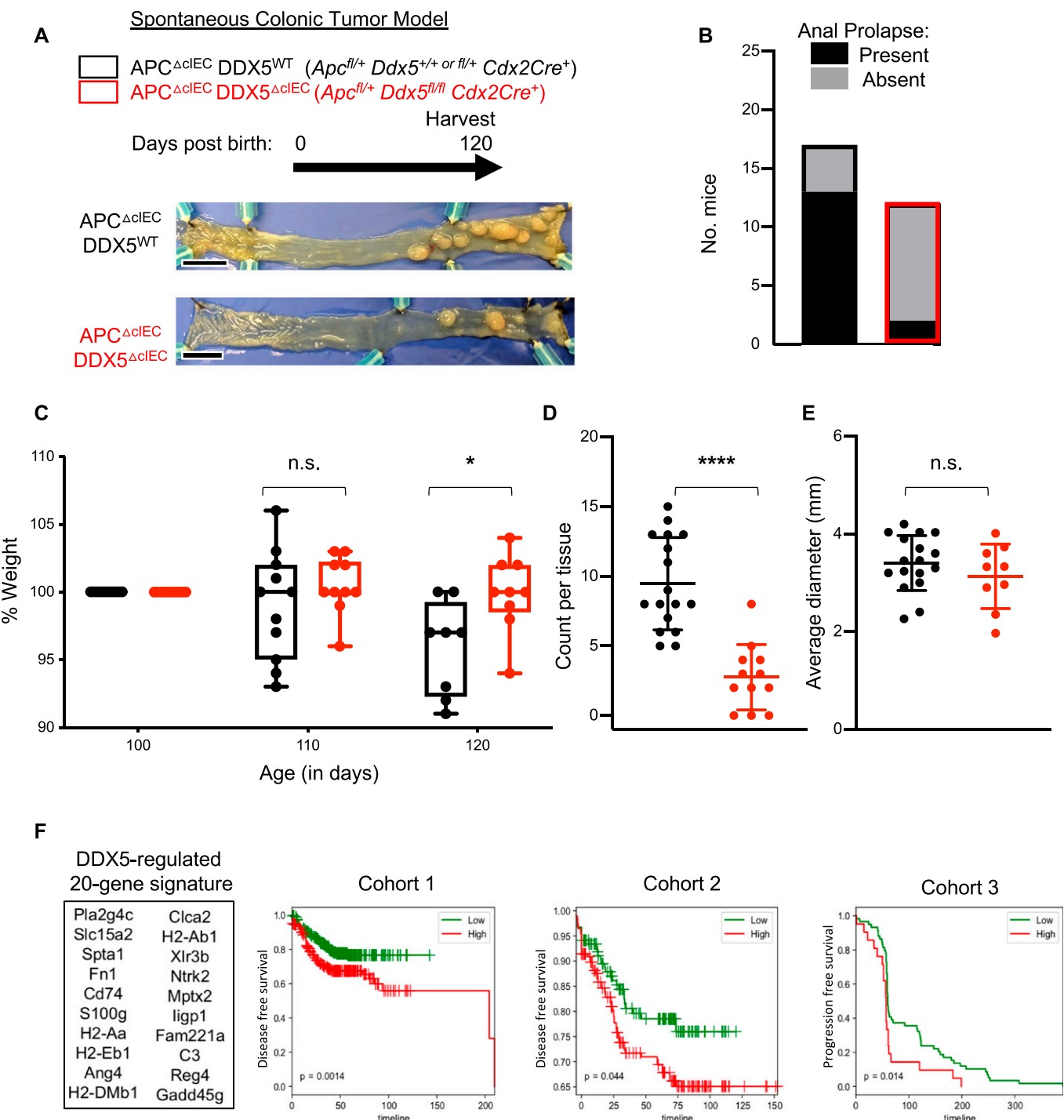

**Figure 2. DDX5 promotes colonic tumorigenesis in *Apc*-mutant mice.**
**(A)** Representative bright-field images of tumor-bearing colons from APC$^{\Delta cIEC}$DDX5$^{WT}$ and APC$^{\Delta cIEC}$DDX5$^{\Delta IEC}$ animals. Scale bar equals 1 cm. **(B)** Anal prolapse incidents recorded in mice described in (A). **(C)** Percent weight change of each mice in (A) on day 110 and 120 compared to day 100. Each dot represents one mouse. Weight change from DDX5-sufficient samples are shown in black (n = 15). Weight change from DDX5 knockouts are shown in red (n = 9). Data shown are means ± SD. *$P$ < 0.05 ($t$ test). **(D)** Colonic tumor counts from APC$^{\Delta cIEC}$DDX5$^{WT}$ ($n$ = 17) and APC$^{\Delta cIEC}$DDX5$^{\Delta cIEC}$ ($n$ = 12) tumor-bearing animals. Each dot represents one mouse. Data shown are means ± SD. **** $P$-value < 0.0001 ($t$ test). **(E)** Average colonic tumor diameter (mm) from APC$^{\Delta cIEC}$DDX5$^{WT}$ ($n$ = 17) and APC$^{\Delta cIEC}$ DDX5$^{\Delta cIEC}$ ($n$ = 9) tumor-bearing animals. Each dot represents one mouse. Data shown are means ± SD. n.s., not significant ($t$ test). **(F)** Expression of the DDX5-dependent colonic gene signature predicts clinical outcome in colorectal cancer patients. Top 20 genes were selected based on criteria listed in the Materials and Methods section. Kaplan–Meier analysis of disease-free survival in cohort 1 (GSE13067, GSE14333, GSE17538, GSE31595, GSE37892, and GSE33113), cohort 2 (GSE87211), and progression-free survival in cohort 3 (GSE5851).
Source data are available for this figure.

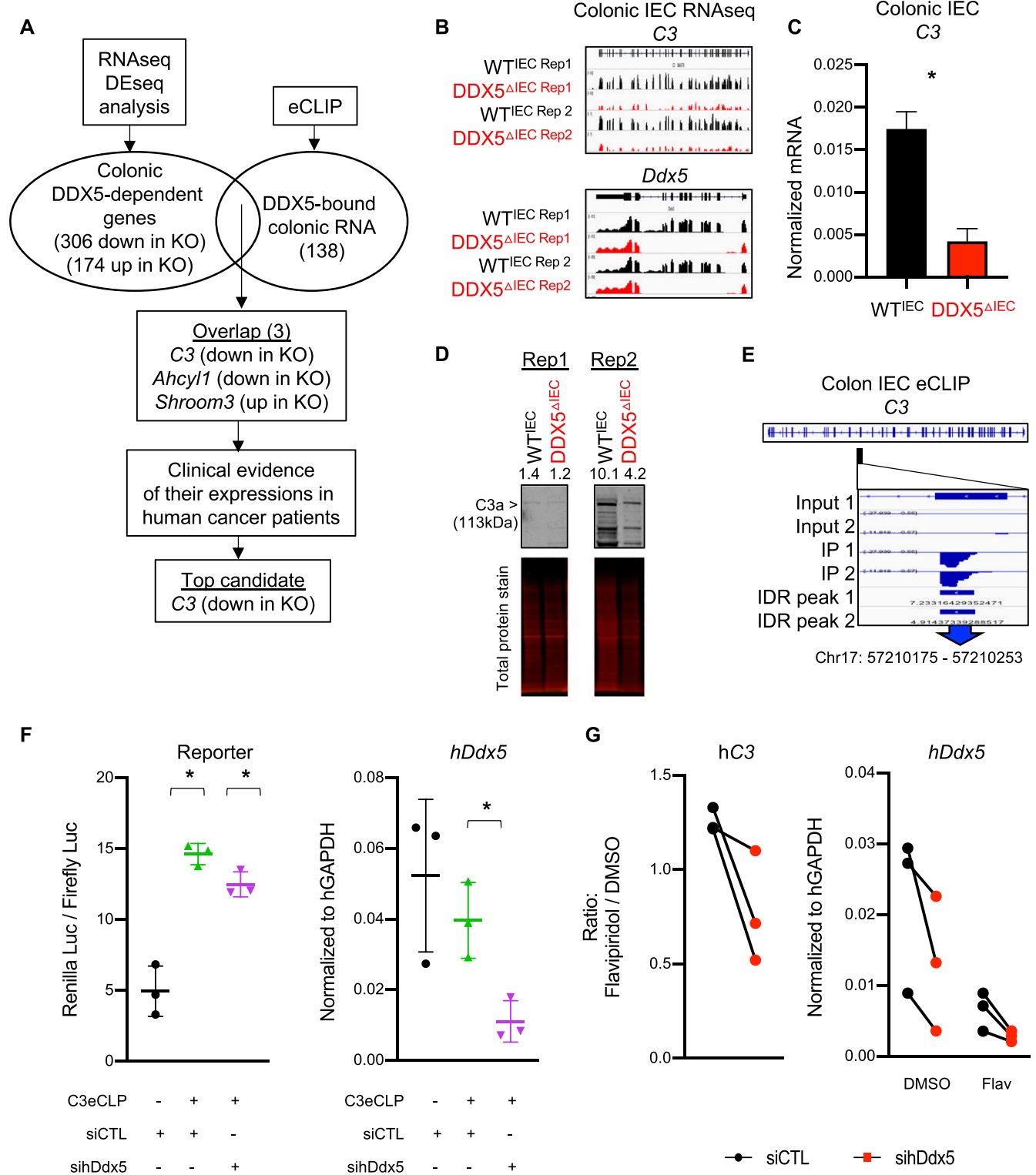

**Figure 3. Epithelial DDX5 binds *C3* RNA to enhance its expression posttranscriptionally.**
**(A)** Workflow to identify DDX5 direct targets in colonic intestinal epithelial cells (IECs) involved in tumorigenesis. **(B)** Integrative Genomics Viewer browser displaying RNA expression at the *C3* and *Ddx5* locus in colonic IECs from two independent pairs of WT[IEC] and DDX5[ΔIEC] littermates. **(C)** qRT-PCR validation of colonic *C3* expression in additional independent pairs of WT[IEC] (n = 3) and DDX5[ΔIEC] (n = 3) animals. Data shown are means ± SD. *P < 0.05 (t test). **(D)** Representative Western analysis of C3 proteins in the colonic IECs from two independent pairs of WT and DDX5-deficient mice. Signal quantification was calculated as signal of C3 over signal of total protein. **(E)** Integrative Genomics Viewer browser displaying the DDX5 binding to *C3* RNAs in WT colonic IECs as defined by eCLIPseq. eCLIPseq was performed on colonic IECs from

demonstrated that knocking out DDX5 in IECs in two models resulted in lower tumor counts. Interestingly, tumors that escaped DDX5 regulation had comparable size as those found in WT animals on day 120, indicating that DDX5 plays a more critical role during tumor initiation and that other regulators can compensate for its loss at the later phase of tumor growth in vivo. We observed that mRNAs encoding other DDX members with structural similarities to DDX5, such as DDX17, are also highly expressed in the intestinal epithelium (Fig 1A). Future studies will be needed to examine whether other DDXs have similar or unique roles in the context of intestinal physiology and pathology.

The characterization of the in vivo DDX5 RNA interactome and regulome uncovered several mechanistic surprises of DDX5 biology. First, we demonstrated that DDX5 preferentially localized to coding regions of RNAs, contributing to RNA stability and/or protein translation of its associated transcripts in mouse IECs. In contrast, previous study in cultured myelogenous leukemia cell line (K562) suggests that DDX5 binding on RNAs is preferentially localized to introns and 5′ UTRs (30, 31, 32, 55, 56). These results suggest that DDX5 binding to RNAs is likely tissue- and cell type specific. We speculate that such specificities may be achieved by DDX5 forming tissue-specific protein complexes with other partners yet to be identified. Future proteomics studies will be needed to uncover the DDX5 protein interactomes in different tissues to address this possibility.

In this study, we focused our mechanistic experiments on two novel targets of DDX5, *C3* and *Fabp1*. Here, we demonstrated that DDX5 binds to and promotes *C3* mRNA stability in colonic IECs. In the small intestine, DDX5 binds to *Fabp1* transcripts, enhancing cytoplasmic RNA levels, and facilitating ribosome engagement to augment the synthesis of FABP1 protein. It remains to be investigated whether the helicase activity of DDX5 is involved in these regulations of epithelial RNA stability and protein translation.

C3 is a potent inducer of the Wnt/β–catenin cascade. DDX5 regulation of C3 uncovered a previously unappreciated role of DDX5 as an upstream regulator of Wnt/β–catenin signaling pathway. Whereas C3 is uniquely expressed in colonic IECs, FABP1 is expressed in the small intestine only. Future studies are needed to investigate the molecular mechanism underlying the region-specific expression of *C3* and *Fabp1* mRNAs observed here (Fig S10A). Regulation of FABP1 by DDX5 revealed a surprising role of DDX5 in intestinal lipid homeostasis. Highly proliferative cells, such as those found in tumor lesions, require large amounts of fatty acid building blocks from exogenous sources and/or de novo synthesis to sustain the building of cell membranes and organelles. For example, previous reports suggest that the up-regulation of acyl-CoA synthetase

long chain family member 4 (ACSL4) promotes tumor cell survival in human colon adenocarcinomas (57), and that fatty acid–binding proteins can channel lipids from surrounding tissues to fuel further tumor growth (58). Future epistasis experiments will be needed to definitively test the contributions of C3, FABP1, and/or other targets acting downstream of DDX5 to promote intestinal inflammation and tumorigenesis. In summary, DDX5 posttranscriptionally orchestrates intestinal RNA programs and drive colitis and intestinal cancers.

# Materials and Methods

### Mice

C57BL/6 wild-type (Stock No: 000664) and *Villin1Cre* (Stock No: 021504) mice were obtained from The Jackson Laboratory. *Ddx5$^{flox}$* mice were obtained from Dr. Frances Fuller-Pace's Laboratory and have been previously described in references 59 and 60. Heterozygous mice were bred to yield 6–8-wk-old *Ddx5$^{+/+}$ Villin1Cre$^{+}$* (subsequently referred to as wild-type, WT$^{IEC}$) and *Ddx5$^{fl/fl}$ Villin1Cre$^{+}$* (referred to as DDX5$^{ΔIEC}$) littermates for experiments related to understanding the role of DDX5 in IECs in both the small intestine and colon. *Apc$^{flox}$* mice were obtained from Dr. Eric Fearon's Laboratory and previously described in reference 44. For our colonic tumor model, *Apc$^{fl/+}$ Ddx5$^{+/+}$ Cdx2Cre$^{+}$* and *Apc$^{fl/+}$ Ddx5$^{fl/+}$ Cdx2Cre$^{+}$* (referred as APC$^{ΔcIEC}$DDX5$^{WT}$), as well as *Apc$^{fl/+}$ Ddx5$^{fl/fl}$ Cdx2Cre$^{+}$* (APC$^{ΔcIEC}$ DDX5$^{ΔcIEC}$) cohoused littermates were used. For our small intestine tumor model, *Apc$^{fl/+}$ Ddx5$^{+/+}$ Villin1Cre$^{+}$* and *Apc$^{fl/+}$ Ddx5$^{fl/+}$ Villin1Cre$^{+}$* (referred as APC$^{ΔIEC}$ DDX5$^{WT}$), and *Apc$^{fl/+}$ Ddx5$^{fl/fl}$ Villin1Cre$^{+}$* (APC$^{ΔIEC}$ DDX5$^{ΔIEC}$) cohoused littermates were used. All animal studies were approved and followed the Institutional Animal Care and Use Guidelines of the University of California San Diego.

### Epithelial cell harvest

Steady-state intestinal epithelial and lamina propria cells were harvested as previously described (61). Briefly, after removing mesenteric fat and Peyer's patches, the proximal 1/3, middle 1/3, and distal 1/3 of the small intestine were designated as the duodenum, jejunum, and ileum, respectively. To isolate IECs, intestine tissues were first incubated in 5 mM EDTA in HBSS containing 1 mM DTT for 20 min at 37°C with shaking, and then incubated in a second wash of 5 mM EDTA in HBSS without DTT for

two independent WT mice. Peaks were called by a cutoff of three for both $\log_{10}$ *P*-values and $\log_2$ (fold changes: immunoprecipitation over input). **(F)** DDX5-binding site on *C3* promotes Renilla luciferase reporter activities in human SW480 cells. Left: reporter activity is calculated as Renilla readings over the constitutive firefly luciferase readings. Results shown are means ± SD of three independent studies. Black: cells transfected with psicheck2 luciferase reporter. Green: cells transfected with psicheck2 luciferase reporter that contains the DDX5-binding site on *C3* and random siRNA. Purple: cells transfected with psicheck2 luciferase reporter that contains DDX5-binding site on *C3* and *Ddx5* siRNA. Right: expression of human DDX5 in SW480 cells were assessed by qRT-PCR and normalized to human GAPDH. SW480 cells under different treatment were indicated as black, green, and purple dots. *$P$ < 0.05 (*t* test). **(G)** RNAi-mediated knockdown of human DDX5 destabilizes *C3* mRNA in Caco-2 cells. Cells were transfected with 100M control (black) or 100M siRNA against human *Ddx5* (red) for 48 h followed by 16 h of incubation with 2 μM flavopiridol. Left: expressions of human *C3* were assessed by qRT-PCR and normalized to DMSO-treated controls. Right: expressions of human *Ddx5* under different treatments were assessed by qRT-PCR and normalized with h*Gapdh*. Results are means of three independent experiments ± SD, *P*-value = 0.06 (*t* test). Source data are available for this figure.

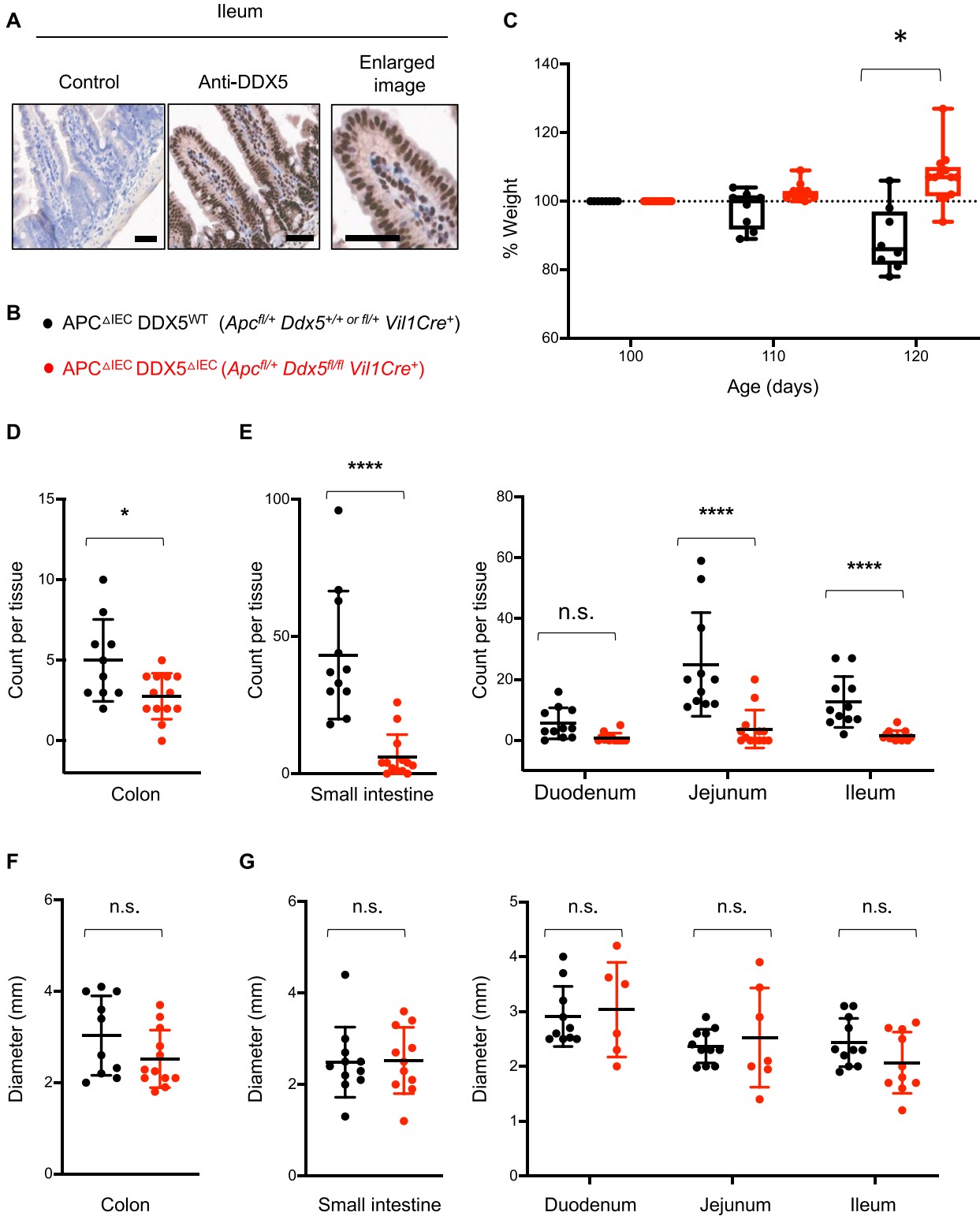

**Figure 4. DDX5 also promotes tumorigenesis in the small intestine.**
**(A)** Representative images from immunohistochemistry analysis of DDX5 in the ileum of WT mice. Enlarged image is shown on the right. Scale bar represents 50 μm. **(B, C, D, E, F, G)** Genotypes of tumor-bearing APC$^{ΔIEC}$DDX5$^{WT}$ and APC$^{ΔIEC}$DDX5$^{ΔcIEC}$ littermates used in (C, D, E, F, G). **(C)** Percent weight change of each mouse in (B) on days 100, 110, and 120. Each dot represents one mouse. Data shown are means ± SD. Each dot represents one mouse. Weight change from DDX5-sufficient samples are shown in black (n = 10). Weight change from DDX5 knockouts are shown in red (n = 13). Data shown are means ± SD. *P < 0.05 (multiple t test). **(D)** Macroscopic tumor counts in the colon. Each dot represents one mouse. Counts from DDX5-sufficient samples are shown in black (n = 12) and counts from DDX5 knockouts are shown in red (n = 13). Data shown

20 min at 37°C with agitation. Suspended cells from the EDTA washes were pooled as "IECs." Colons were processed similarly.

## Histology and IHC

Ileal and colonic tissues were fixed overnight in 10% formalin at room temperature. Paraffin-embedded tissues were sectioned into 5-$\mu$m slices, stained with H&E, periodic acid-Schiff, or IHC (see Table S4 for antibody information). Briefly, paraffin sections were de-paraffinized and rehydrated with TBST washes between each step (Tris-buffered saline, pH 7.8, with 0.1% Tween-20). Sections were blocked first against endogenous peroxidases (immersed for 30 min in 0.3% $H_2O_2$) and then blocked against endogenous biotin using unlabeled streptavidin and excess free biotin. Antigen retrieval was induced by heating the slide for 5 min twice in 10 mM sodium citrate buffer, pH 6.0, followed by 20 min of cooling. Finally, the sections were blocked against nonspecific hydrophobic interactions with 1% BSA/TBST. Staining was then performed with either the negative control IgG antibody or anti-DDX5 and anti-Ki67 (1:100) antibodies overnight in a humid chamber at 4°C. The next day, the sections were washed with TBST and then sequentially overlaid with biotinylated goat anti-rabbit (111-065-045; Jackson ImmunoResearch) at 1:500, followed by HRP-labeled streptavidin (16-030-084; Jackson ImmunoResearch) at 1:500. Substrate was then overlaid (AEC from Vector labs following directions) for 30 min followed by nuclear counterstain with Mayer's hematoxylin. Images were acquired using the AT2 Aperio Scan Scope (UCSD Moores Cancer Center Histology Core).

## Western blot

For whole cell lysates, cells were lysed in 25 mM Tris, pH 8.0, 100 mM NaCl, and 0.5% NP40 with protease inhibitors for 30 min on ice. Samples were spun down at 14,000$g$ for 15 min, and soluble protein lysates were harvested. The NE-PER kit (Thermo Fisher Scientific) was used for cytoplasmic and nuclear fractionation studies. 30–50 $\mu$g protein was loaded on each lane. Blots were blocked in Odyssey Blocking Buffer (LI-COR) and probed for the desired proteins. After incubation with respective IRDye secondary antibody (LI-COR), infrared signals on each blot were measured on the LI-COR Odyssey CLX. The primary antibodies used in this study are listed in Table S4.

## cDNA synthesis and qRT-PCR

Total RNA was extracted with the RNeasy Plus Kit (QIAGEN) and reverse-transcribed using iScriptЇ Select cDNA Synthesis Kit (Bio-Rad). Real-time RT-PCR was performed using iTaq Universal SYBR Green Supermix (Bio-Rad). For IECs and tumor RNA expression, data were normalized to *Gapdh*. Primers were designed using Primer-BLAST to span across splice junctions, resulting in PCR amplicons that span at least one intron. Primer sequences are listed in Table S8.

## RNAseq

Ribosome-depleted RNAs were used to prepare sequencing libraries. 100-bp paired-end sequencing was performed on an Illumina HiSeq4000 by the Institute of Genomic Medicine (IGM) at the University of California San Diego. Each sample yielded ~30–40 million reads. Paired-end reads were aligned to the mouse mm10 genome with the STAR aligner version 2.6.1a ([62]) using the parameters: "--outFilterMultimapNmax 20 –alignSJoverhangMin 8 –alignSJDBoverhangMin 1 –outFilterMismatchNmax 999 –outFilterMismatchNoverReadLmax 0.04 –alignIntronMin 20 –alignIntronMax 1000000 –alignMatesGapMax 1000000." Uniquely mapped reads overlapping with exons were counted using featureCounts ([63]) for each gene in the GENCODE.vM19 annotation. Differential expression analysis was performed using DESeq2 (v1.18.1 package) ([64]), including a covariate in the design matrix to account for differences in harvest batch/time points. Regularized logarithm (rlog) transformation of the read counts of each gene was carried out using DESeq2. Pathway analysis was performed on differentially expressed protein coding genes with minimal counts of 10, $\log_2$ fold change cutoffs of ≥0.5 or ≤−0.5, and *P*-values < 0.05 using Gene Ontology (http://www.geneontology.org/) where all expressed genes in the specific cell type were set as background.

Gene set enrichment analysis was carried out using the pre-ranked mode of the gene set enrichment analysis software with default settings ([55], [65]). The gene list from DEseq2 was ranked by calculating a rank score of each gene as −$\log_{10}$(*P*-value) × sign ($\log_2$ FoldChange), in which FoldChange is the fold change of expression in DDX5$^{\Delta IEC}$ over those found in WT$^{IEC}$.

## Enhanced cross-linked immunoprecipitation (eCLIPseq)

eCLIPseq analysis was performed as previously described ([50]). For IEC eCLIPseq, the cells were isolated from two 8–10-wk-old wild-type (C57BL/6) female mice, as described above, and 50 million cells from each mouse were used in the two biological replicates. The cells were subjected to UV-mediated cross-linking, lysis, and treatment with limiting amounts of RNases, followed by IP of the DDX5-containing RNA complexes. RNA fragments protected from RNase digestion were subjected to RNA linker ligation, reverse-transcription, and DNA linker ligation to generate eCLIPseq libraries for high-throughput Illumina sequencing.

are means ± SD. *$P$ < 0.05 ($t$ test). **(E)** Left: total macroscopic tumor counts in the small intestine. Right: Macroscopic tumor counts in different segments of the small intestine. Each dot represents one mouse. Counts from DDX5-sufficient samples are shown in black (n = 12) and counts from DDX5 knockouts are shown in red (n = 13). Data shown are means ± SD. n.s., not significant. ****$P$ < 0.0001 (multiple $t$ test). **(F)** Average tumor diameters in the colon. Each dot represents one mouse. Diameters from DDX5-sufficient samples are shown in black (n = 12) and diameters from DDX5 knockouts are shown in red (n = 13). Data shown are means ± SD. n.s., not significant ($t$ test). **(G)** Left: Average tumor diameters in the small intestine. Right: average tumor diameters in different segments of the small intestine. Each dot represents one mouse. Diameters from DDX5-sufficient samples are shown in black (n = 12) and diameters from DDX5 knockouts are shown in red (n = 13). Data shown are means ± SD. n.s., not significant ($t$ test).
Source data are available for this figure.

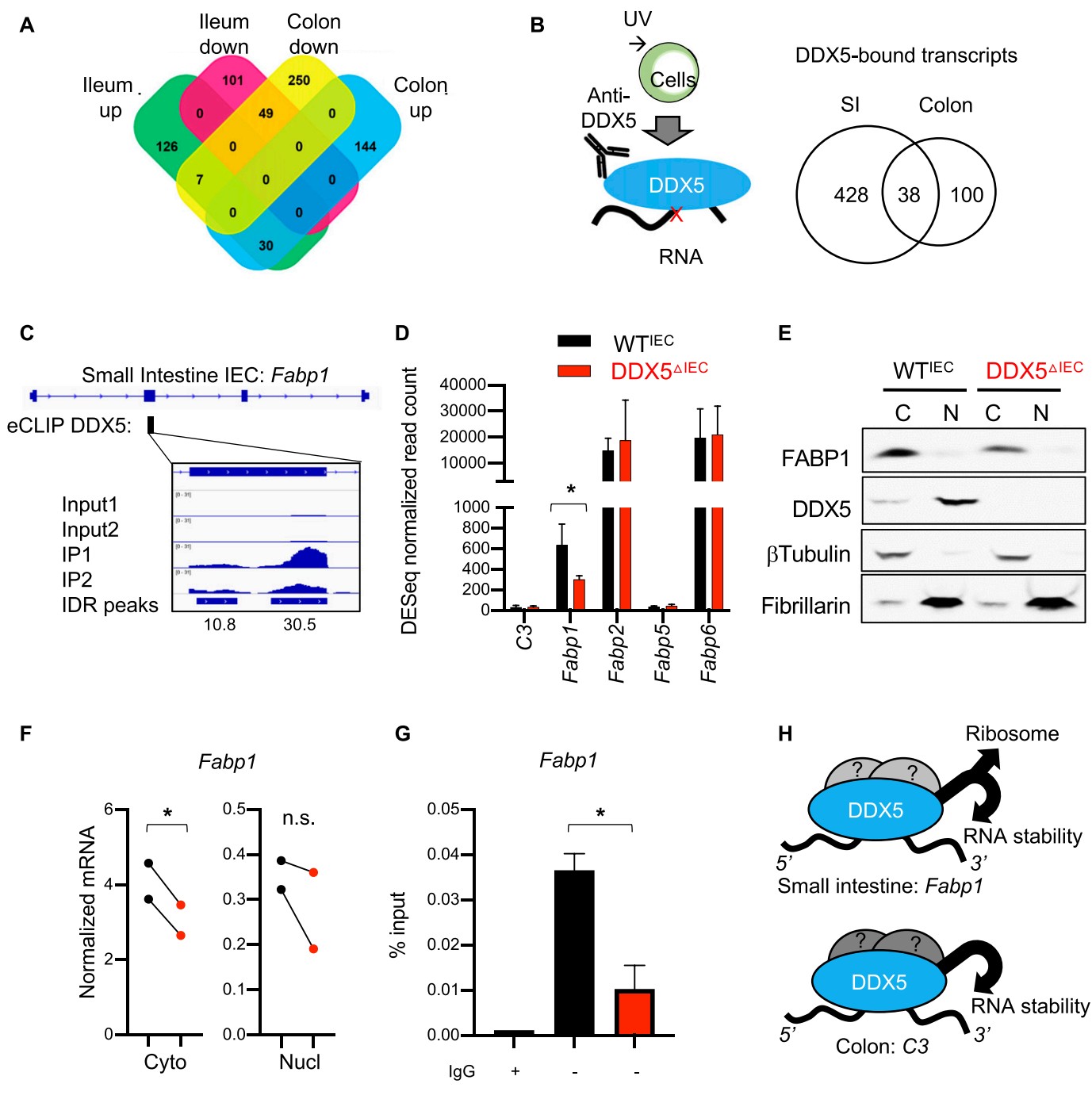

**Figure 5. DDX5 regulates overlapping and distinct RNA programs in the small intestine and colon.**
**(A)** Venn diagram of the overlapping and distinct DDX5-dependent transcripts from the ileum and colon defined as $\log_2$ fold change of ≥0.5 or ≤−0.5 and *P*-value < 0.05. RNAseq was performed on two independent pairs of cohoused DDX5$^{\Delta IEC}$ over WT$^{IEC}$ littermates. **(B)** Venn diagram showing the overlapping and distinct DDX5-bound transcripts in the small intestine and colonic intestinal epithelial cells (IECs). eCLIPseq was performed on small intestine IECs from two independent WT mice. Peaks were called by a cutoff of three for both $\log_{10}$ *P*-values and $\log_2$ (fold changes: immunoprecipitation over input). **(C)** Integrative Genomics Viewer browser displaying DDX5 binding on the fatty acid-binding protein 1 (*Fabp1*) locus as defined by eCLIPseq. **(D)** Normalized RNAseq read counts of transcripts encoding members of the FABP family in ileal IECs from WT$^{IEC}$ and DDX5-deficient mice. *P < 0.05 (DESeq). **(E)** Representative Western blots for FABP1, DDX5, *β*-tubulin, and fibrillarin in cytoplasmic (C) and nuclear (N) extracts of small intestine IECs from WT and DDX5$^{\Delta IEC}$ mice. Experiments were repeated three times using independent biological samples with similar results. **(F)** RNAs from the nuclear and cytoplasmic fractions of small intestine IECs harvested from WT and DDX5$^{\Delta IEC}$ mice were evaluated by qRT-PCR for *Fabp1*. Each dot represents one mouse. This experiment was repeated on two pairs of independent samples. *P < 0.05 (*t* test). **(G)** Engagement of *Fabp1* mRNA with ribosome RPL10A in small intestine IECs. Results are means of two independent experiments ± SD. *P < 0.05 (*t* test). **(H)** Working model: DDX5 posttranscriptionally regulates the expression of tissue-specific oncogenic RNAs in IECs.
Source data are available for this figure.

Peak regions were defined using CLIPper first on the IP sample (https://github.com/YeoLab/clipper/wiki/CLIPper-Home). Enrichment was calculated using both the IP and input samples. $\log_2$ fold change was calculated as eCLIPseq reads normalized for read depth over normalized reads found at each peak region in the size-matched input sample. ENCODE Irreproducible Discovery Rate analysis was performed on two independent biological replicates of IECs. Peaks were ranked using the entropy formula, Pi*log(Pi/Qi)/$\log_2$, where Pi is the probability of an eCLIPseq read at that position and Qi is the probability of input reads at that position. Results were filtered using cutoffs of three for both $\log_{10}$ P-values and $\log_2$ fold changes, respectively, to define a set of true peaks normalized above their respective size-matched input background signal.

### ChIP

ChIP was carried out as described previously (56). Briefly, 20 million intestinal IECs were fixed with 1% formaldehyde at room temperature for 10 min and quenched with 125 mM glycine for 5 min at room temperature. All buffer compositions were described in reference 56. Nuclear lysates were sonicated with a Bioruptor (Branson Sonifier Cell Disruptor 185) at 4°C using the output setting at 4 for 10 cycles of 30 s on and 30 s off. 30 $\mu$g of chromatin was used per IP. Chromatin was diluted 10× in ChIP dilution buffer supplemented with proteinase inhibitor. 5% of the total chromatin used per IP reaction was saved as input samples. 5 $\mu$g of antibody was added per 30 $\mu$g chromatin per IP reaction and incubated overnight at 4°C. The immune complexes were then incubated with 30 $\mu$l of Dynabeads Protein G (10004D; Thermo Fisher Scientific) for 4 h at 4°C on rotation. After washes, protein–DNA complexes were eluted from the beads by adding 200 $\mu$l of elution buffer and incubating the beads at 65°C for 15 min with constant shaking at 1,000$g$. Eluted samples were incubated at 37°C for 30 min with 1 $\mu$l of DNase and protease-free RNase A (10 mg/ml, EN0531; Thermo Fisher Scientific). DNA and protein cross-links were reversed by adding 8 $\mu$l of i NaCL and 2 $\mu$l of proteinase K solution (20 mg/ml, AM2546; Thermo Fisher Scientific) by overnight incubation at 65°C under constant shaking. Chromatin was isolated using QIAGEN QIAquick PCR Purification Kit (28104) and eluted in 40 $\mu$l elution buffer. Input samples were diluted five times to make a 1% input control. The ChIP signals were calculated as follows: Adjusted input = Ct (Input) – 6.644. ChIP signal = 100 × Power (2; average of adjusted Input-Ct value ChIP sample). All ChIP qPCR primers are listed in Table S8.

### Ribosome pull-down assay

Small intestine or colonic IECs were lysed in polysome extraction buffer (10 mM Hepes, pH 7.4, 150 mM KCl, 5 mM MgCl$_2$, 1% NP40, 2 mM dithiothreitol, 80 U/ml RNaseOUT, 100 $\mu$g/ml cycloheximide, and protease inhibitors). Cell extracts were subject to anti-ribosome IP overnight with 2 $\mu$g anti-RPL10A antibodies (Abcam) and harvested in Protein G magnetic Dynabeads (Invitrogen) as described previously (66). cDNAs were synthesized from purified RNA using Superscript III (Invitrogen). Level of RPL10A associated transcripts in pull-down was calculated as fraction of input for each sample.

### DSS-induced colitis

Mice were provided 2% (wt/vol) DSS (160110; MP Biomedicals) in their drinking water for 7 d, followed by 7 d of access to regular drinking water. Mice were monitored daily for their weight and tissues were harvested on day 15 post-DSS treatment. Pathology scoring of distal colon from DSS-challenged mice was performed blind by JE Hernandez following previously published guidelines (67), including parameters for inflammatory infiltrate, crypt density, crypt hyperplasia, muscle thickening, and submucosal inflammation.

### Intestine organoid cultures

Isolated colonic crypts were embedded in Corning Matrigel Matrix Corning Matrigel GFR Membrane Matrix (CB40230C; Thermo Fisher Scientific) and seeded onto pre-warmed 24-well plates (CytoOne) and overlaid with conditioned media as described in reference 68. The organoid images were acquired using fluorescence microscopy (11350119; Thermo Fisher Scientific).

### RNAi in human IECs

Caco-2 cell line was cultured in 1× DMEM/F12 media (Gibco, Life Technologies). The media were supplemented with 1× 10% FBS (Gibco, Life Technologies), 1 mM sodium pyruvate (Gibco, Life Technologies) and 1% penicillin streptomycin (Gibco, Life Technologies). Cells were plated on a 24-well plate at 500 liters/well at 2 × 10$^5$ cells/ml 1 d before transfection. 50 $\mu$M human DDX5 siRNA pool (Cat. no. D-003774-02, D-003774-03, D-003774-04, D-003774-17; GE Healthcare Dharmacon, see Table S9) or scramble siRNA pool (Cat. no. D-001206-14-05; GE Healthcare Dharmacon, see Table S9) were mixed with Opti-MEM medium and Lipofectamine 3000 (L3000001; Invitrogen) reagent according to the manufacturer's protocol. Solutions were vortexed and incubated for 5 min at room temperature to allow the formation of siRNA–lipid complex. 50 $\mu$l of transfection/siRNA (final concentration of 100 nM) mixture was added to the well and incubated at 37°C. Transcription inhibitor, flavopiridol, was purchased from Sigma-Aldrich (F3055). After incubation for 48 h, the cells were treated with DMSO or flavopiridol (2 $\mu$M) and were collected 16 h posttreatment. RNA extraction and qRT-PCR were carried out as described above.

### Luciferase assay

The psiCheck2 construct containing dual *Renilla* and Firefly luciferase reporters was purchased from Promega (Promega). The DDX5-bound sequence on *Fabp1* was cloned into a multiple cloning site located downstream of the Renilla translational stop codon. SW480 cells were cultured in 1× DMEM/F12 media (Gibco, Life Technologies). 1× 10% FBS (Gibco, Life Technologies), 1 mM sodium pyruvate (Gibco, Life Technologies) and 1% penicillin streptomycin (Gibco, Life Technologies) were added. The cells were plated on a 96-well plate at 0.5 × 10$^5$ cells/ml 1 d before transfection. 100 nM of siCTL or the human DDX5 siRNA pool were introduced to SW480 cells using Lipofectamine 3000 (L3000001; Invitrogen) in Opti-MEM medium according to the manufacturer's protocol. The transfection mixture was incubated at room temperature for 5 min. 10 $\mu$l of the

transfection mixture was added to each well and incubated at 37°C for 24 h. 1 μg of psiCheck2 luciferase reporter plasmids were transfected with Lipofectamine 3000 and 1 μl P3000 enhancer reagent in the Opti-MEM medium. After 24 h, the cell lysates were used to measure both *Renilla* and firefly activities using the Dual-Luciferase Reporter Assay System (Promega) according to the manufacturer's instructions.

### Kaplan–Meier analysis

Top 20 DDX5-dependent genes were selected based on two criteria: $log_2$ FoldChange < −1 (down-regulated in DDX5-deficient colonic IECs) and adjusted *P*-value < 0.05. Gene expression values were normalized according to a modified Z-score approach centered around StepMiner threshold (formula = (expr − SThr)/3*stddev). A composite gene expression score is computed based on linear combination of normalized and z-scored scaled expression values. StepMiner algorithm (69) is used to classify the final score into high and low values. Kaplan–Meier analysis of disease-free survival in two independent cohorts reveal strong (Pooled gene expression omibus [GEO]: GSE13067, GSE14333, GSE17538, GSE31595, GSE37892, GSE33113, n = 555, *P* = 0.0014; GSE87211, n = 351, *P* = 0.044) association between 20 down-regulated gene signatures and worse outcome. Progression-free survival analysis were performed on the mCRC GSE5851 dataset (*P* = 0.014).

### Accession numbers

The accession numbers for the small intestine and colon eCLIPseq results reported in this article are available on GEO (GSE124023). The IEC RNAseq datasets are available on GEO (GSE123881).

### Statistical analysis

All values are presented as mean ± SD. Significant differences were evaluated using GraphPad Prism 8 software. The *t* test was used to determine significant differences between two groups. A two-tailed *P*-value of <0.05 was considered statistically significant in all experiments.

# Supplementary Information

# Acknowledgements

We thank Eric Fearon for sharing the APC conditional mutant mice previously described in reference 44. We thank Frances Fuller-Pace for sharing the DDX5 conditional mutant mice previously described in reference 60. We thank Giuseppe Di Caro for Apc mouse colony breeding advice. We thank Jean YJ Wang for critical reading of this manuscript. N Abbasi, T Long, Y Li, BS Cho, E Ma, PR Patel, and WJM Huang are partially funded by the Edward Mallinckrodt, Jr. Foundation and the National Institutes of Health (NIH) (R01 GM124494 to WJM Huang). JE Hernandez is funded by NIH (R01 GM124494-03S1 to WJM Huang). BA Yee and GW Yeo are partially funded by NIH (R01 HG004659 and U41 HG009889 to GW Yeo). D Sahoo is partially supported by NIH (R01 GM138385, R00 CA151673 to D Sahoo, and UG3 TR002968), and Padres Pedal the Cause/Rady Children's Hospital Translational PEDIATRIC Cancer Research Award (#PTC2017), Padres Pedal the Cause/C3 Collaborative Translational Cancer Research Award (San Diego NCI Cancer Centers Council [C3] #PTC2017) and the Leonna M Helmsley Charitable Trust. IM Sayed and S Das are partially funded by NIH (R01 DK107585 to S Das and UG3 TR002968), the DiaComp Pilot and Feasibility award to S Das (Augusta University) and the Leonna M Helmsley Charitable Trust. P Ghosh is supported by NIH (R01 AI141630 to P Ghosh and UG3 TR002968), and the Leonna M Helmsley Charitable Trust. RNAseq was conducted at the IGM Genomics Center, University of California San Diego, with support from NIH (S10 OD026929). We thank the UC San Diego HUMANOID Core and the Moores Cancer Center Histology Core Tissue Technology Shared Resource for the IHC analysis, supported by NIH (P30 CA23100).

## Author Contributions

N Abbasi: formal analysis, validation, visualization, and writing—original draft, review, and editing.
T Long: data curation, formal analysis, validation, and writing—review and editing.
Y Li: data curation, software, formal analysis, and methodology.
BA Yee: data curation, software, and formal analysis.
BS Cho: data curation, formal analysis, and writing—review and editing.
JE Hernandez: data curation, formal analysis, and writing—review and editing.
E Ma: formal analysis.
PR Patel: formal analysis and writing—review and editing.
D Sahoo: formal analysis and visualization.
IM Sayed: conceptualization and writing—review and editing.
N Varki: data curation and formal analysis.
S Das: conceptualization, supervision, and writing—review and editing.
P Ghosh: conceptualization, supervision, funding acquisition, visualization, and writing—review and editing.
GW Yeo: conceptualization, supervision, funding acquisition, and writing—original draft, review, and editing.
WJM Huang: conceptualization, data curation, formal analysis, supervision, funding acquisition, visualization, and writing—original draft, review, and editing.

## Conflict of Interest Statement

GW Yeo is co-founder, member of the Board of Directors, on the Science Advisory Board, equity holder, and paid consultant for Locanabio and Eclipse BioInnovations. GW Yeo is a visiting professor at the National University of Singapore. GW Yeo's interests have been reviewed and approved by the University of California San Diego in accordance with its conflict of interest policies. All other authors declare that they have no conflict of interest.

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
