## [Reviewer comments · Life Science Alliance]

1st Editorial Decision

5 May 2020

DDX5 promotes oncogene C3 and FABP1 expressions and drives intestinal inflammation and tumorigenesis

Nazia Abbasi, Tianyun Long, Yuxin Li, Brian Yee, Benjamin Cho, Juan Hernandez, Evelyn Ma, Parth Patel, Debashis Sahoo, Ibrahim Sayed, Nissi Varki, Soumita Das, Pradipta Ghosh, Gene Yeo, Wendy Jia Men Huang
DOI: 10.26508/lsa/202000772

Corresponding author(s): Dr. Wendy Jia Men Huang (University of California San Diego)

Review timeline:

Submission Date:	2020-05-12
Editorial Decision:	2020-05-12
Revision Received:	2020-07-04
Editorial Decision:	2020-08-04
Revision Received:	2020-08-07
Accepted:	2020-08-07

Transaction Report:

Please note that the manuscript was previously reviewed at another journal and the reports were taken into account in the decision-making process at Life Science Alliance. Since the original reviews are not subject to Life Science Alliance's transparent review process policy, the reports and author response cannot be published.

Re: Life Science Alliance manuscript #LSA-2020-00772-T

Wendy Jia Men Huang
University of California San Diego

Dear Dr. Huang,

Thank you for submitting your manuscript entitled "DDX5 targets tissue-specific RNAs to promote intestine tumorigenesis" to Life Science Alliance. The manuscript was assessed by expert reviewers at another journal, and the editors transferred those reports to us with your permission.

The two reviewers who evaluated your study at the other journal did not raise strong technical concerns, but would have expected further genetic evidence for the proposed link between reduced APC-model-dependent tumor burden when combined with a DDX5 deficient background and altered expression levels of C3 in the colon and FABP1 in the small intestine, respectively.

We see value in describing the DDX5-APC mutant mouse model phenotype and providing transcriptomic insight into the changes occurring, and would thus like to invite you to submit a revised version of your manuscript to us based on the reviewer reports already at hand. We would expect a point-by-point response to all concerns raised and accordingly additional experimentation, clarification, and changes to manuscript text and data representation. The requested further genetic evidence does not need to get provided, but the proposed link to C3/FABP1 mis-regulation should get substantiated by other additional data, such as monitoring the inflammatory status of the mice in response to challenges. It would be also good to add clinical evidence showing the correlation of DDX5 and C3/FABP1 expression in human intestinal/colorectal cancers in response to rev#2. You already outlined prior to transferring your study that adding these data is feasible despite the current pandemic. We thus think the requested revision can be performed in a relatively normal time frame, but we'd be happy to extend the revision time should this be necessary.

To upload the revised version of your manuscript, please log in to your account: <https://lsa.msubmit.net/cgi-bin/main.plex>
You will be guided to complete the submission of your revised manuscript and to fill in all necessary information. Please get in touch in case you do not know or remember your login name.

Thank you for this interesting contribution to Life Science Alliance. We are looking forward to receiving your revised manuscript.

Sincerely,

B. MANUSCRIPT ORGANIZATION AND FORMATTING:

We encourage our authors to provide original source data, particularly uncropped/processed electrophoretic blots and spreadsheets for the main figures of the manuscript. If you would like to add source data, we would welcome one PDF/Excel file per figure for this information. These files will be linked online as supplementary "Source Data" files.

2nd Editorial Decision

4 August 2020

RE: Life Science Alliance Manuscript #LSA-2020-00772-TR

Dr. Wendy Jia Men Huang
University of California San Diego
9500 Gilman Drive MC0651
George Palade Laboratories, Room 321B
La Jolla 92093

Dear Dr. Huang,

Thank you for submitting your revised manuscript entitled "DDX5 promotes oncogene C3 and FABP1 expressions and drives intestinal inflammation and tumorigenesis". We would be happy to publish your paper in Life Science Alliance pending final revisions necessary to meet our formatting guidelines.

- We note the deletion of 3 panels in figs 1 and 3A in response to points #7, #8, #12 of ref 1. While publication is not contingent on this, we repeat the suggestion that rather than delete the data it could actually be presented in more compelling graph form
- please add a conflict of interest statement to your main manuscript text
- Please remove panel A (in the figure and the figure legend) for Figures: S3, S6, S9, S14, S15
- Please make sure that the origin box matches the magnification in your Figures (Figure S7C)
- please use the [10 author names, et al.] format in your references (i.e. limit the author names to the first 10)

To upload the final version of your manuscript, please log in to your account: <https://lsa.msubmit.net/cgi-bin/main.plex>

A. FINAL FILES:

-- Summary blurb (enter in submission system): A short text summarizing in a single sentence the study (max. 200 characters including spaces). This text is used in conjunction with the titles of papers, hence should be informative and complementary

to the title. It should describe the context and significance of the findings for a general readership; it should be written in the present tense and refer to the work in the third person. Author names should not be mentioned.

B. MANUSCRIPT ORGANIZATION AND FORMATTING:

We encourage our authors to provide original source data, particularly uncropped/-processed electrophoretic blots and spreadsheets for the main figures of the manuscript. If you would like to add source data, we would welcome one PDF/Excel file per figure for this information. These files will be linked online as supplementary "Source Data" files.

Sincerely,

Reilly Lorenz
Editorial Office Life Science Alliance
Meyerohofstr. 1
69117 Heidelberg, Germany
t +49 6221 8891 414
e contact@life-science-alliance.org
www.life-science-alliance.org

3rd Editorial Decision

7 August 2020

RE: Life Science Alliance Manuscript #LSA-2020-00772-TRR

Dr. Wendy Jia Men Huang
University of California San Diego
9500 Gilman Drive MC0651
George Palade Laboratories, Room 321B
La Jolla 92093

Dear Dr. Huang,

Thank you for submitting your Research Article entitled "DDX5 promotes oncogene C3 and FABP1 expressions and drives intestinal inflammation and tumorigenesis". It is a pleasure to let you know that your manuscript is now accepted for publication in Life Science Alliance. Congratulations on this interesting work.

*****IMPORTANT: If you will be unreachable at any time, please provide us with the email address of an alternate author. Failure to respond to routine queries may lead to unavoidable delays in publication.*****

DISTRIBUTION OF MATERIALS:

Again, congratulations on a very nice paper. I hope you found the review process to be constructive and are pleased with how the manuscript was handled editorially. We look forward to future exciting submissions from your lab.

Sincerely,

Reilly Lorenz
Editorial Office Life Science Alliance
Meyerohofstr. 1

Life Science Alliance

Life Science Alliance - Peer Review Process File

69117 Heidelberg, Germany
t +49 6221 8891 414
e contact@life-science-alliance.org
www.life-science-alliance.org